# Mathematical Modeling of Vaccines That Prevent SARS-CoV-2 Transmission

**DOI:** 10.3390/v13101921

**Published:** 2021-09-24

**Authors:** David A. Swan, Ashish Goyal, Chloe Bracis, Mia Moore, Elizabeth Krantz, Elizabeth Brown, Fabian Cardozo-Ojeda, Daniel B. Reeves, Fei Gao, Peter B. Gilbert, Lawrence Corey, Myron S. Cohen, Holly Janes, Dobromir Dimitrov, Joshua T. Schiffer

**Affiliations:** 1Vaccine and Infectious Disease Division, Fred Hutchinson Cancer Research Center, Seattle, WA 98109, USA; dswan@fredhutch.org (D.A.S.); agoyal@fredhutch.org (A.G.); jrmoore@fredhutch.org (M.M.); ekrantz@fredhutch.org (E.K.); erbrown@fredhutch.org (E.B.); ecojeda@fredhutch.org (F.C.-O.); dreeves@fredhutch.org (D.B.R.); fgao@fredhutch.org (F.G.); pgilbert@scharp.org (P.B.G.); lcorey@fredhutch.org (L.C.); hjanes@fredhutch.org (H.J.); ddimitro@scharp.org (D.D.); 2TIMC-IMAG/BCM, Université Grenoble Alpes, 38000 Grenoble, France; chloe@bunchmountain.com; 3Public Health Sciences Division, Fred Hutchinson Cancer Research Center, Seattle, WA 98109, USA; 4Department of Biostatistics, University of Washington, Seattle, WA 98195, USA; 5Department of Medicine, University of Washington, Seattle, WA 98195, USA; 6Department of Laboratory Medicine, University of Washington, Seattle, WA 98195, USA; 7Clinical Research Division, Fred Hutchinson Cancer Research Center, Seattle, WA 98109, USA; 8Institute of Global Health and Infectious Diseases, University of North Carolina at Chapel Hill, Chapel Hill, NC 27599, USA; myron_cohen@med.unc.edu; 9Department of Applied Mathematics, University of Washington, Seattle, WA 98195, USA

**Keywords:** SARS-CoV-2, vaccines, mathematical modeling, viral dynamics

## Abstract

SARS-CoV-2 vaccine clinical trials assess efficacy against disease (VE_DIS_), the ability to block symptomatic COVID-19. They only partially discriminate whether VE_DIS_ is mediated by preventing infection completely, which is defined as detection of virus in the airways (VE_SUSC_), or by preventing symptoms despite infection (VE_SYMP_). Vaccine efficacy against transmissibility given infection (VE_INF_), the decrease in secondary transmissions from infected vaccine recipients, is also not measured. Using mathematical modeling of data from King County Washington, we demonstrate that if the Moderna (mRNA-1273QS) and Pfizer-BioNTech (BNT162b2) vaccines, which demonstrated VE_DIS_ > 90% in clinical trials, mediate VE_DIS_ by VE_SUSC_, then a limited fourth epidemic wave of infections with the highly infectious B.1.1.7 variant would have been predicted in spring 2021 assuming rapid vaccine roll out. If high VE_DIS_ is explained by VE_SYMP_, then high VE_INF_ would have also been necessary to limit the extent of this fourth wave. Vaccines which completely protect against infection or secondary transmission also substantially lower the number of people who must be vaccinated before the herd immunity threshold is reached. The limited extent of the fourth wave suggests that the vaccines have either high VE_SUSC_ or both high VE_SYMP_ and high VE_INF_ against B.1.1.7. Finally, using a separate intra-host mathematical model of viral kinetics, we demonstrate that a 0.6 log vaccine-mediated reduction in average peak viral load might be sufficient to achieve 50% VE_INF,_ which suggests that human challenge studies with a relatively low number of infected participants could be employed to estimate all three vaccine efficacy metrics.

## 1. Introduction

The endpoint for SARS-CoV-2 vaccine efficacy trials targeting licensure is vaccine efficacy against disease (VE_DIS_), which is defined by a reduction in symptomatic disease, confirmed with polymerase chain reaction (PCR) testing for viral RNA, in vaccine recipients relative to placebo recipients [1,2]. The FDA benchmark for licensure is a point estimate of VE_DIS_ > 50% with a lower alpha-adjusted 95% confidence limit exceeding 30% [3]. The two mRNA vaccines which have been widely used across the United States showed high levels of protection (>90%) in trials and upon follow up [4,5,6].

Once VE_DIS_ is established and a vaccine is licensed, mathematical modeling is useful for projecting a roll out strategy that affords maximal reductions in deaths and cases, and to prevent the need for future lockdowns [7,8,9]. Yet VE_DIS_ does not provide sufficient information to fully inform these models. High VE_DIS_ is determined by a combination of two distinct phenomena which were only partially captured in these trials: vaccine efficacy against susceptibility (VE_SUSC_), which is defined as the vaccine-induced reduction in the rate of infection as evidenced by detection of virus by PCR, and vaccine efficacy against symptoms (VE_SYMP_) which is defined as the reduction in the presence of symptoms conditional on infection under vaccine versus placebo (Table 1, Figure 1) [1,2,10]. If a vaccine mediates VE_DIS_ primarily through reduction in symptoms, the extent to which people remain asymptomatic despite infection because of receiving the vaccine, but can still transmit the virus, remains unknown. A vaccine that achieves high VE_DIS_ via VE_SYMP_ could theoretically contribute less to overall herd immunity than a vaccine that achieves high VE_DIS_ via VE_SUSC_, as the former may not block ongoing chains of transmission from vaccine recipients.

Another vaccine effect, efficacy against transmissibility given infection (VE_INF_) is defined as reduction in secondary transmissions from either symptomatic or asymptomatic infected vaccine versus placebo recipients and could also have significant effects on the trajectory of viral epidemics [11]. Reduced VE_INF_ anticipates that symptomatic breakthrough infections in vaccine recipients may be associated with fewer secondary transmissions than in placebo recipients, and that people who develop asymptomatic rather than symptomatic infection due to vaccination (VE_SYMP_) may also be less likely to transmit. This latter observation would be expected if a vaccine mediates reduction in both symptoms and secondary transmission potential by lowering the quantity of viral shedding [12]. While high VE_DIS_ guarantees a high likelihood of individual benefit, protection of unvaccinated members of the population will also depend on VE_SUSC_ and VE_INF_, as well as the velocity of a vaccination rollout program [8,13,14].

The inability to fully discriminate VE_SUSC_ from VE_SYMP_, and to directly measure VE_INF_ in the current slate of promising vaccines limits our ability to forecast vaccine impacts in the population. Specifically, there is uncertainty regarding the proportion of vaccinated people required to achieve the herd immunity threshold, where the effective reproductive number (R_eff_), given a certain degree of continued social distancing, is maintained below 1 [15]. It is similarly challenging to optimize vaccine allocation to different sectors of the population, particularly when vaccine supply is limited. For instance, it may be best to target a vaccine with high VE_SUSC_ or VE_INF_, which breaks secondary chains of transmission, towards essential workers and young people [8]. Alternatively, a vaccine with high VE_SYMP_ but limited effects on secondary transmission may be best prioritized towards populations with highest risk of severe disease, such as the elderly.

Several possible methods exist to estimate VE_INF_. One is to measure secondary attack rate among household contacts of infected vaccine recipients versus infected placebo recipients [16,17]. Alternatively, cluster-randomized trials can assess for indirect protection of unvaccinated persons in vaccinated versus unvaccinated communities [18]. While both trial designs are attractive, they have high operational complexity and need to be implemented and completed rapidly to impact the course of the pandemic.

Another option is to use a viral load metric as a surrogate endpoint. VE_INF_ is likely to be mediated at least in part via a reduction in viral load among recipients of vaccine versus placebo, particularly early during pre-symptomatic or asymptomatic infection when nasal and saliva viral loads are highest [19,20,21,22]. It is possible that VE_SYMP_ is also driven by viral load reduction, though it has yet to be proven beyond association whether any specific viral load metric predicts development of symptoms or severe COVID-19 [23]. Moreover, only a few studies captured critical early peak viral load kinetics, and in too few people to perform correlate analyses [20,24,25]. Viral load in infected vaccine versus placebo recipients could be measured in large clinical trials in which enrolled participants undergo frequent self-sampling after enrollment, or in smaller highly controlled human challenge studies [26].

Recent data provides some insight that VE_SUSC_ had favorably high values in current SARS-CoV-2 vaccines through May 2021. Data from clinical trials of the Moderna (mRNA-1273QS) and Janssen vaccines suggests 60–70% protection against PCR confirmed infection after a single vaccine dose [4,27]. Observational studies of health care workers, community members and long-term care residents who were followed with serial testing suggest significant levels of protection following full vaccination [28,29,30,31]. Serial assessment for infection in British households suggests high vaccine induced protection against SARS-CoV-2, including the dominant variant of concern B.1.1.7 or alpha during the fourth wave of infection in spring 2021. More, recently there has been some erosion in VE_SUSC_ associated with the predominance of B.1.167.2 or delta [6]. While not providing a precise estimate of VE_SUSC_, these data suggest that at least some observed efficacy against disease in clinical trials is mediated by complete protection against infection.

While a precise estimate of VE_INF_ is also lacking, mounting evidence suggests that the current widely deployed mRNA vaccines have this effect against the baseline SARS-CoV-2 variants as well as B.1.1.7. A meta-analysis suggests that asymptomatic infection is associated with an 85% relatively lower secondary attack rate than symptomatic or pre-symptomatic infection [32]. Moreover, viral load among infected people who received the Pfizer (BNT162b2) vaccine was observed to be 0.5–2.0 logs lower than in unvaccinated cohorts [33], though these studies did not selectively capture the critical pre-symptomatic phase of symptomatic infection when viral load and transmissibility are highest [21,34,35]. Moreover, prior to B.1.167.2 predominance, secondary attack rates among household contacts of vaccinated health care workers were 50% lower relative to unvaccinated controls [36].

Here we use a mathematical modeling approach using data from King County Washington to demonstrate the potential effects of VE_INF_ at the population level given multiple vaccine profiles. In contrast to several existing models of vaccine prioritization [37,38], the model accounts for the likely need for recurrent lockdowns if cases and hospitalizations exceed a certain threshold. It also accounts for a variant like the B.1.17 viral variant with higher infectiousness [39,40]. We next estimate reduction in peak viral load required to achieve various VE_INF_.

## 2. Materials and Methods

### 2.1. Overview

In Section 3.1, Section 3.2, Section 3.3, Section 3.4, Section 3.5, Section 3.6 and Section 3.7 of the Results, we used an epidemiologic transmission mathematical model of COVID-19 in King County Washington to project the theoretical impact of different vaccine efficacy profiles (which are defined in Table 1) on infections, deaths, need for further lockdown and timing of herd immunity thresholds in 2021. This model was previously employed to demonstrate the importance of rapid vaccination rates to limit the severity of a fourth wave in King County Washington [14]. In the next section of the paper, we employed an intra-host transmission model [22,35,41] to evaluate whether reduction in peak viral load could serve as a potential correlative endpoint for VE_INF_, which could then inform human viral challenge study designs which might provide actionable vaccine efficacy estimates within relevant timeframes for the pandemic.

### 2.2. King County Transmission Model

We modified a previously developed deterministic compartment model [37] which captures the epidemic dynamics in King County, WA (population = 2.25 million people) between January 2020 and January 2021 and projects the trajectory of the local pandemic through the end of 2021 in the absence and presence of vaccines. This model was selected based on pre-existing parameterization and its accessibility for testing the impact of vaccination campaigns.

Vaccination is simulated with a starting date of 1 January 2021. Our model stratifies the population by age (0–19 years, 20–49 years, 50–69 years, and 70+ years), infection status (uninfected, exposed, pre-symptomatic, symptomatic, asymptomatic, hospitalized, dead and recovered as in Figure 2), clinical status (undiagnosed, diagnosed when asymptomatic or symptomatic) and vaccination status. Full equations and parameter descriptions are in the Supplement. Parameters and their values, as well as King County specific demographic data are in Appendix A.

We assumed that 20% of infections are asymptomatic and that asymptomatic people are as infectious as symptomatic individuals but missing the highly infectious pre-symptomatic phase. As a result, the relative infectiousness of individuals who never develop symptoms is assumed to be 56% of the overall infectiousness of individuals who develop symptomatic COVID-19. This conservative estimate falls between the 35% relative infectiousness estimated in a recent review based on 79 studies [42] and the estimate of 75% suggested by the CDC [43].

The forces of infection, representing the risk of the susceptible individuals to acquire infection (transition from susceptible to exposed), are differentiated by age of the susceptible individual, the contact matrix (proportion of contacts with each age group), infection and treatment status (asymptomatic, pre-symptomatic, symptomatic, diagnosed, and hospitalized cases) of the infected contacts as described in the Supplement.

The model is parameterized with local demographic and contact data from King County, WA and calibrated to local case and mortality data using transmission parameters ranges informed from published sources [34,44,45,46]. The calibration is described in the Supplement.

Based on current Washington state policies, unless otherwise noted, we assume that in the absence of a vaccine, numbers of cases and hospitalizations fluctuate due to the community response to the epidemic [47,48]. When the number of new infections remains below a certain threshold, physical distancing measures are assumed to relax, allowing greater contact between susceptible and infected people. The effective reproductive number (R_eff_) may eventually exceed one and cases will start growing in number. Ultimately a threshold may be surpassed that necessitates re-enforcement of physical distancing restrictions: R_eff_ drops below one and cases contract.

A critical parameter in the model is the social distancing metric which estimates the amount of potential infection contacts between members of the population. This parameter is intended to capture physical contact reduction due to physical distancing policies, but also the decreased number of transmission contacts due to masking. The parameter varies between 0, which represents pre-pandemic levels of interactivity, and 1, which represents complete physical distancing with no interactivity. The parameter is implemented as a proportional reduction in the infectiousness component of the force of infection term between infected and susceptible people.

We arrived at values for this parameter in each age cohort by calibrating the model to retrospectively observed infection, hospitalization and death data through the end of January 2021, and then allowed to vary prospectively in accordance with Washington state policy regarding future lockdowns. Our benchmarks for increasing physical distancing to 0.6 was when two-week average number of cases exceeded 350 per 100,000 [49]. After 1 January 2021, we allowed relaxation of the parameter to 0.3 when the two-week average number of cases fell below 100 per 100,000. For elderly populations, we assume greater restrictions to 0.8 and lessened relaxation to 0.4. This approach reproduces the waves of infection which have defined the United States and King County epidemics to date.

Finally, following 1 January, we assume the presence of a variant similar to B.1.1.7 giving the circulating virus 55% greater infectiousness than the variants which predominated during the first three waves [50].

### 2.3. Vaccine Simulations in King County, Washington

We sought to define the effect of different vaccine profiles on incident and cumulative infections and deaths as well as requirements for achieving the herd immunity threshold when R_eff_ < 1. We considered several vaccine efficacy profiles as described in the Results with different efficacies as defined in Table 1. Implementation of these efficacies is described in the Supplement.

We consider scenarios in which VE_SUSC_, VE_SYMP_, and VE_INF_ each have either low (10%), medium (50%) or high (90%) efficacy. Each possible parameter combination allows for 3^3^ (27) vaccine scenarios. Five VE_SUSC_ and VE_SYMP_ combinations (5 × 3 or 15 scenarios when considering the 3 values of VE_INF_): VE_SUSC_ = 90%/VE_SYMP_ = 10%; VE_SUSC_ = 10%/VE_SYMP_ = 90%; VE_SUSC_ = 90%/VE_SYMP_ = 50%; VE_SUSC_ = 50%/VE_SYMP_ = 90%; VE_SUSC_ = 90%/VE_SYMP_ = 90%) would be compatible with current projections for the Moderna and Pfizer mRNA vaccines against B.1.1.7 which had estimated VE_DIS_ = 95% and 90%, respectively [4,5]. Three combinations or nine scenarios (VE_SUSC_ = 50%/VE_SYMP_ = 50%; VE_SUSC_ = 50%/VE_SYMP_ = 10%; VE_SUSC_ = 10%/VE_SYMP_ = 50%) would be realistic if there is a relative decrease in VE_DIS_ due to new viral variants as appears to be occurring with the now predominant B.1.617.2 variant of concern [6]. These lower vaccine estimates may also be relevant for other vaccines in development, or after a single dose of the Moderna or Pfizer product. One combination or three scenarios (VE_SUSC_ = 10%/VE_SYMP_ = 10%) which would not meet licensure requirements are included as controls to independently assess the effect of increasing VE_INF_.

To reflect the rate of vaccination in King County, we initially assumed 10,000 vaccinations per day with the goal of covering 90% of the adult population. We also simulated lower vaccination rates (5000 per day) to capture vaccination campaigns in other settings. In our simulations, both susceptible and recovered persons were vaccine eligible. We assumed that the vaccine start date represented the timing of the second shot for the mRNA vaccines such that efficacy accrues at the defined time of vaccination. We imputed no loss of vaccine efficacy over time.

In keeping with state vaccination programs, we initially assumed disproportionate initial targeting of the cohorts aged >70 (80% of vaccines with 20% to those older than 20 years old). We also imputed a slow relaxation of social distancing during the vaccination program when cases remained below a certain threshold.

### 2.4. SARS-CoV-2 Inra-Host and Transmissions Models

In Results Section 3.7, we used a separate set of models to estimate the viral load reduction required to achieve clinically relevant values for VE_INF_ in a clinical trial. We employed an intra-host model describing SARS-CoV-2 infection from our previous study to generate viral loads to assess transmission risk [22]. The viral load generating model is included in the Supplement with references to prior data fitting. We also employed our previously described model linking transmitter viral load with probability of transmission [22,35], which was validated against published data including variability in number of infections generated by individuals and distributions in observed serial intervals [21,51,52]. As previously described, the model output predicts a transmission dose response curve which captures probability of transmission given an exposure viral load [35,41]. The details of this model are described in the Supplement and in Appendix A.

We simulated the impact of the vaccination in this model by assuming that a vaccine generates a certain number of SARS-CoV-2 specific acquired immune cells that are ready to proliferate and quickly eliminate the ongoing infection as a necessary condition to lower peak viral load in infections such as SARS-CoV-2 with rapid initial growth kinetics [53]. We thereby modified and simplified our prior intra-host model to:$$\frac{\mathrm{dS}}{\mathrm{dt}}=-\beta \mathrm{VS}$$
$$\frac{\mathrm{dI}}{\mathrm{dt}}=\beta \mathrm{VS}-{\delta \mathrm{II}}^{k}-\mathrm{mEI}$$
$$\frac{\mathrm{dV}}{\mathrm{dt}}=\pi I-\gamma V$$
$$\frac{\mathrm{dE}}{\mathrm{dt}}=\frac{\omega \mathrm{IE}}{I+I_{50}}$$

$I_{50}$ denotes the level of infected cell that allows proliferation of immune cells at 50% maximal. We assume it to be 10 cells/mL. We further fix ω=2 days^−1^cells^−1^ [54] and m=0.01 days^−1^cells^−1^. The latter parameter value is scalable to variable E which is not directly measured experimentally and induces its dynamic effect (mEI) via relative changes despite the absolute value E at any point in time not being identifiable. While we assume vaccine induced immunity to be cell-mediated, we previously demonstrated that we can generate equivalent viral kinetics assuming humoral immunity as a cause for reduced viral load [22].

To simulate different vaccine efficacies, we assume a different starting condition of parameter E (E_0_) that leads to predictable reductions in peak viral load. We simulated 1000 vaccine recipients and 1000 placebo recipients under each condition, and then assessed the relative reduction in transmissions to estimate VE_INF_ as in Table 1.

## 3. Results

### 3.1. High Projected Incidence of SARS-CoV-2 Infections, Deaths and Lockdown in King County Washington in 2021 without Vaccination

Our King County Washington model [37] accurately recapitulated the three prior waves of daily diagnosed cases (Appendix A), daily hospitalizations (Appendix A), daily deaths (Appendix A), age-stratified diagnosed cases (Appendix A), age-stratified hospitalizations (Appendix A), age-stratified deaths (Appendix A), cumulative diagnosed cases (Appendix A), cumulative hospitalizations (Appendix A), and cumulative deaths (Appendix A) through December 2020.

Extending beyond the calibration period, due to the higher infectiousness of the B.1.1.7 variant and lack of sufficient prior infection to reach the herd immunity threshold, we projected a substantial fourth wave in the absence of vaccination (black line Figure 3, wave number in top row) with a peak exceeding 5000 daily infections in early June (Figure 3a) and 20 daily deaths (Figure 3b). We also anticipated the need to re-enforce physical distancing between May and November 2021 to achieve 40% interactivity relative to pre-pandemic levels (Figure 3c) in order to lower R_eff_ below 1 (Figure 3d). At the end of this fourth wave, we forecasted that more than 30% of the population had been infected, including more than 3500 total deaths, most of which occurred during the fourth wave (Figure 3a,b, bottom row).

### 3.2. Moderate Vaccine Efficacy against Infection or High Vaccine Efficacy against Secondary Transmission as a Mitigator against a Fourth Wave of a Variant Similar to B.1.1.7 SARS-CoV-2 Infections, Deaths, and Lockdown in 2021

We considered scenarios in which elderly cohorts were vaccinated first at rates comparable to those in King County thus far and VE_DIS_ was mediated mostly by VE_SUSC_ rather than VE_SYMP_ (VE_SYMP_ = 10%). Vaccines with high protection against infection (VE_SUSC_ = 90%) resulted in substantial reductions in fourth wave peak (Figure 3 top row) and cumulative (Figure 3 bottom row) infections (Figure 3a) and deaths (Figure 3b). All vaccines with VE_INF_ = 90% prevented a rapidly expanding fourth wave of infections and deaths and eliminated the need for lockdown during summer of 2021 (Figure 3c), while maintaining R_eff_ less than 1 (Figure 3d). For VE_SUSC_ = 90% vaccines compatible with Moderna and Pfizer results, increasing VE_INF_ from 10% to 90% had a slight additional effect on reducing infections and deaths.

All vaccines with at least 50% VE_SUSC_ or 90% VE_INF_ lead to a reduction of at least 200,000 infections and 1000 deaths since the start of the vaccination period. A vaccine with VE_SUSC_ = 10%, VE_INF_ = 10% and VE_SYMP_ = 10% was predicted to slightly delay and blunt the peak of infections and deaths (Figure 3 top row), with a moderate reduction in these outcomes (Figure 3 bottom row) and a requirement for a five-month phase of increased physical distancing (Figure 3c). A vaccine with VE_SUSC_ = 50%, VE_INF_ = 10% and VE_SYMP_ = 10% necessitated a three-month period of increased physical distancing to suppress the fourth wave.

### 3.3. High Vaccine Efficacy against Secondary Transmission as a Requirement for Prevention of a Fourth Wave of a Variant Similar to B.1.1.7 SARS-CoV-2 Cases, Deaths, and Lockdown in 2021 for Vaccines with High Efficacy against Symptoms but Low Efficacy against Infection

We next considered a scenario in which VE_DIS_ was mediated mostly by reduction in symptoms (VE_SYMP_ = 50 or 90% with low VE_SUSC_ = 10%) with initial vaccine prioritization to the elderly and equivalent daily vaccination rates. For all conditions with VE_INF_ = 90%, we observed a relative decrease in infections (Figure 4a) and a substantial relative decrease in deaths (Figure 4b) with a delayed but protracted fourth wave (Figure 4, top row) and a lower cumulative incidence of both outcomes (Figure 4, bottom row). There was a substantial decrease in deaths but not infections associated with increasing VE_SYMP_ from 50% to 90% when VE_INF_ = 90%. Both scenarios were associated with no further need for reactive lockdown (Figure 4c).

Under the high VE_SYMP_, low VE_INF_ scenario, which could be compatible with the Moderna and Pfizer vaccine trial results, a fourth protracted wave peaking at >5000 daily infections and >5 daily deaths lasting from May through October, 2021 occurred with a period of reactive lockdown between July and October (Figure 4c,d). For moderate VE_INF_ (50%) and lower VE_INF_ (10%), we observed a beneficial effect of increased VE_SYMP_ (Figure 4) with a reduction in deaths at high (90%,) versus moderate (50%) VE_SYMP_.

### 3.4. Ranges of Possible Outcomes under All Scenarios Compatible with Moderna and Pfizer Clinical Trial Results

We explored the impact of varying VE_INF_ under the entire range of plausible vaccine scenarios with VE_DIS_ = 90% which could be compatible with the Moderna and Pfizer vaccine clinical trial results. We generated heat maps for total post-vaccine diagnosed cases (Figure 5a) and deaths (Figure 5b) and identified that for scenarios when VE_DIS_ = 90% is mediated entirely by VE_SYMP_ (90%), increasing VE_INF_ from 10% to 90% resulted in substantial further reductions in post-vaccine diagnosed cases (>20,000) and deaths (>200). When VE_DIS_ = 90% was mediated entirely (90%) by VE_SUSC_, then increasing VE_INF_ from 10% to 90% resulted in lower reductions in diagnosed cases (>5000) and deaths (>50).

### 3.5. Vaccine Efficacy as a Determinant of Fourth Wave Severity Assuming Low Vaccination Rate

The distribution and acceptability of vaccines to the public has varied across the United States and the world. We therefore simulated scenarios assuming half (5000 vaccines/day, Figure 6) the vaccination rate. We assumed VE_SYMP_ = 90% such that all six considered scenarios had efficacies compatible with the Moderna and Pfizer clinical trial results. At this slower roll out, a fourth wave of infections (Figure 6a) and deaths (Figure 6b) occurred among all scenarios. The peak (Figure 6, top row) was somewhat blunted under scenarios with VE_SUSC_ = 90% or VE_INF_ = 90% with >200,000 fewer cumulative infections and >1500 fewer deaths relative to no vaccination (Figure 6, bottom rows), indicating that VE_INF_ would take on added importance under less optimal roll out scenarios with low VE_SUSC_. All scenarios permitted a severe enough wave to necessitate another round of required lockdown to stem the severity of the fourth wave (Figure 6c,d).

### 3.6. Variant Infectiousness, Vaccine Efficacy and Vaccination Rate as Key Determinants of Number of Infections Prior to Attainment of the Herd Immunity Threshold

We next considered how different vaccine scenarios might impact the timing of achieving the herd immunity threshold when R_eff_ <1 as well as the cumulative number of infected and vaccinated people when this threshold is reached (Figure 7). We considered three variables: the infectiousness of the viral variant (baseline or 55% increased as with the B.1.1.7 variant), the vaccination rate (5000 per day versus 10,000 day), and the vaccine efficacy profile. For all vaccines, we assumed VE_SYMP_ = 90% such that all simulations were compatible with Moderna and Pfizer clinical trial results. Finally, for these simulations of the fourth wave, we assumed baseline social distancing of 0.2 with no reactive physical distancing because altering the social distancing metric over time confounds this result. (A further reduction in social distancing would increase the herd immunity threshold.)

For the baseline variant in the absence of vaccination (Figure 7a,b top row), the peak number of diagnosed cases exceeded 3000 per day (>12,000 new infections per day) in early August, 2021. The herd immunity threshold was reached after more than 35% of the population had been infected (Figure 7a,b middle row). By the end of 2021, 55% of the population was projected to have been infected.

For a variant similar to the B.1.1.7 variant in the absence of vaccination (Figure 7c,d top row), the number of diagnosed cases peaked at greater than 4500 per day (>18000 infections per day) in early June 2021. The herd immunity threshold was reached after >55% of the population had been infected (Figure 7c,d middle row). By the end of 2021 >90% of the population had been infected.

With a vaccination rate of 5000 per day assuming the baseline variant, the peak number of daily diagnosed cases was profoundly diminished to fewer than 500 per day assuming any vaccine efficacy with either VE_SUSC_ = 50% or 90%, or VE_INF_ = 90% (Figure 7a, top row). The herd immunity threshold was surpassed for most vaccines in mid-April when less than 25% of the population had been vaccinated (Figure 7a, bottom row). Vaccinations rather than new infections (Figure 7a, middle row) contributed to reaching the herd immunity threshold and fewer than 15% of the population were infected by the end of 2021. Even for a weak vaccine with VE_SUSC_ = 10%, VE_INF_ = 10% and VE_SYMP_ = 90% (Figure 7a, bottom row), a blunted fourth wave was projected, and the herd immunity threshold was reached in late July when approximately 45% of the population had received the vaccine. New infections contributed somewhat to attaining the herd immunity threshold under this scenario in which greater than 30% of the population was infected by the end of the fourth wave.

With a vaccination rate of 10,000 per day assuming the baseline variant, no significant fourth wave occurred even with VE_SUSC_ = 10%, VE_INF_ = 10% and VE_SYMP_ = 90% (Figure 7b, top and middle row): the herd immunity threshold was surpassed for this scenario in mid-June when approximately 65% of the population had been vaccinated (Figure 7c, bottom row).The herd immunity threshold was surpassed for most other vaccines in early March when greater than 20% of the population had been vaccinated (Figure 7b, bottom row). Under these scenarios, fewer than 20% of the population were infected by the end of 2021.

With a vaccination rate of 5000 per day assuming a variant like B.1.1.7, the number of daily diagnosed cases was only slightly blunted and delayed relative to no vaccination, even assuming high vaccine efficacy with VE_SUSC_ = 90% and/or VE_INF_ = 90% (Figure 7c, top row). By blunting the cumulative number of infections, vaccination was projected to slightly delay the time to herd immunity threshold in all cases (Figure 7c, middle and bottom rows). The herd immunity threshold was surpassed for all vaccines in June when more than 35% of the population were vaccinated (Figure 7c, bottom row). Vaccinations and new infections (Figure 7c, middle row) contributed to reaching the herd immunity threshold under all scenarios. The vaccine efficacy profile had a substantial impact on the ratio of vaccinated to infected people at the time of herd immunity threshold. Vaccines with high VE_SUSC_ (90%) permitted the fewest cumulative infections (approximately 50% of the population by the end of 2021) while vaccines with high VE_INF_ (90%) but moderate or low VE_SUSC_ (50% or 10%) allowed a higher number of incident infections (Figure 7c, middle row) (60–70% of the population by the end of 2021).

With a vaccination rate of 10,000 per day assuming a variant like B.1.1.7, the number of daily diagnosed cases was projected to be highly dependent on the vaccine efficacy profile (Figure 7d, top row). Vaccines with high VE_SUSC_ (90%) permitted the fewest cumulative infections (approximately 15% by the end of 2021), while vaccines with high VE_INF_ (90%) but moderate or low VE_SUSC_ (50% or 10%) allowed a slightly higher number of infections (Figure 7d, middle row). The herd immunity threshold was surpassed for these vaccines in June when more than 60% of the population were vaccinated and only 10% had been infected (Figure 7d, middle and bottom row). A vaccine with VE_SUSC_ = 50% and VE_INF_ = 10% allowed a severe but delayed fourth wave which ultimately infected approximately 50% of the population.

### 3.7. Small Reduction in Peak Viral Load Required for Lowering VE_INF_

The above results suggest that the potential severity of subsequent SARS-CoV-2 waves can only be projected with accurate estimates for VE_SUSC_, VE_SYMP_ and VE_INF_ among relevant vaccines, as well as rates of vaccine rollout and infectiousness of future viral variants. It is therefore a priority to identify the true values for these parameters.

Based on experience from multiple viruses that show that exposure dose predicts transmission [55,56], we hypothesize that VE_INF_ is likely to be mediated by a reduction in viral load among infected people (Appendix A). We therefore employed an intra-host model described in the Methods and Supplement that links SARS-CoV-2 viral load dynamics in an infected person with transmission potential. This model is entirely separate from the King County model in Section 3.1, Section 3.2, Section 3.3, Section 3.4, Section 3.5 and Section 3.6 and is intended to link individual viral load with transmission dynamics.

We next considered methods to estimate VE_INF_ using viral load as a potential surrogate. We established a relationship between the initial number of tissue resident immune cells and peak viral load (Appendix A) during individual simulated infections. We then assumed vaccination of 1000 people in which vaccine recipients generated a certain number of these immune cells while placebo recipients did not. By estimating the reduction in the number of transmissions, we were then able to estimate VE_INF_ for each vaccine. The model predicted a saturating relationship between reduction in peak viral load and VE_INF_: a 0.6 log or fourfold reduction in peak viral load resulted in VE_INF_ = 50% and a 2.5 log or ~300-fold reduction resulted in VE_INF_ = 90% (Appendix A).

## 4. Discussion

An optimal vaccine program would prevent the maximum numbers of cases and deaths without the need for further lockdown periods. The first component of such a program was the testing and licensing of vaccines that provide protection from symptomatic disease (VE_DIS_). Initial data from the Pfizer and Moderna trials and follow up cohort studies suggest that these products have greater than 90% VE_DIS_ against the original and B.1.1.7 variants [4,5]. The second step is to consider the proportion of the population that will need to be vaccinated to surpass the herd immunity threshold. This threshold will depend critically on indirect effects that protect unvaccinated members of the population. Indirect effects occur when VE_DIS_ is mediated by protection against infection (VE_SUSC_), rather than protection against symptoms despite infection (VE_SYMP_) but may also be augmented by a vaccine product with high protection against secondary transmission despite infection (VE_INF_). Given rapid enough roll out, our results suggest that vaccines with either high VE_SUSC_ or high VE_INF,_ plus moderate VE_SYMP_ would have limited a severe fourth wave of cases and deaths related to a variant like the B.1.1.7 variant in 2021. With slower vaccine rollout, relative improvements in VE_SUSC_ or VE_INF_ may have led to massive reductions in numbers of infections and deaths.

VE_SUSC_ can only be partially discriminated from VE_SYMP_ in most clinical trials to date using serologic assays which may miss infection due to waning humoral responses [57]. Moreover, VE_INF_ was not directly assessed. VE_SUSC_, VE_SYMP_ and VE_INF_ are particularly challenging to measure, leaving policy makers with incomplete information for projecting the impact of a given vaccine even after trial results are available. While evidence from observational studies [28,29,30] suggest that the mRNA COVID-19 vaccines retained high VE_SUSC_ in early 2021 when B.1.1.7 predominated, this challenge remains relevant as new viral variants continue to emerge across the globe, which may result in reductions in some but not all components of vaccine efficacy.

We identified that under any scenario in which VE_SUSC_ is low, a vaccine with VE_INF_ >50% adds substantial protection at the population level. A vaccine with this profile would exert maximal benefit if rolled out quickly enough. If VE_SYMP_ had driven observed trial results, then high VE_INF_ would have been vital for preventing cases and deaths. In scenarios where a fourth spring wave was inevitable, such as caused by highly contagious new variants [50,58,59] or a slow vaccine rollout, VE_INF_ could potentially have delayed or blunted the peak number of cases and deaths, thereby preventing the need for reinforcement of physical distancing measures while also preventing many deaths.

In reality, the fourth wave in King County Washington peaked at ~400 diagnosed cases in early May (which in our model equates to fewer than 2000 infections per day) and two to three deaths per day [44]. This result suggests that high VE_SUSC_ (Figure 3), or a combination of both high VE_SYMP_ and high VE_INF_ (Figure 4) against the B.1.1.7 variant is likely to explain observed data. In other words, the Moderna and Pfizer vaccines either completely prevented most infections or did not prevent infection but did mostly eliminate both symptoms and secondary transmissions from those infected despite receiving the vaccine.

It remains an urgent research priority to continually update estimates for VE_SUSC_, VE_SYMP_ and VE_INF_ for vaccines, particularly against variant B.1.167.2 and future variants of concern which may exhibit different levels of immune evasion [60,61]. Studies which attempt to directly measure secondary infections in households [62,63], or to assess the degree of protection afforded to unvaccinated members of communities with partial vaccination relative to communities with less vaccination, would potentially be useful. They would need to be performed quickly to obtain actionable results and may suffer from confounding relative to controlled clinical trials, however.

Our second analysis suggested that peak viral load could serve as a surrogate endpoint for secondary transmission and allow for rapid, complementary studies. We previously estimated the relationship between viral load and transmission probability for SARS-CoV-2 [35]. The emergent transmission response curve has a similar sigmoidal shape to empirically-derived curves for SARS-CoV-1 in a controlled set of murine experiments [64] and SARS-CoV-2 in non-human primates [65], and resembled the relationship between quantitative viral PCR and probability of culture positivity in humans infected with SARS-CoV-2 [66].

As a first step, it is necessary to formally test the hypothesis that exposure viral load is predictive of transmission risk. A valid viral load surrogate cannot currently be inferred from human cohorts as the exposure viral load is rarely documented between transmission pairs, though formal surrogate endpoint analysis will ultimately be necessary if sufficient data emerges. Animal models of infection are ideal for this purpose and the necessary transmission dose could be inferred with a relatively small number of non-human primates or mice [64,65].

Human studies using reduction in peak viral load or viral area under the curve as correlates for reduction in VE_INF_ could take one of two forms. The first would involve prospective nasal sampling of virus in all enrolled participants with virologic endpoints compared between those who become infected in vaccine and placebo arms. An ideal trial population would be university students due to their high incidence rate and low overall infection morbidity. The advantages of this approach would be real-world validation of biologic vaccine effects in which participants experience natural variability in potentially critical factors such as viral exposure dose, time between vaccination and infection, and route of transmission. The relationship between viral load and symptoms would also be clarified with this study design. Challenges would be operational including large samples size and a massive number of prospective samples.

Human challenge studies are a potentially rapid method to directly measure VE_SUSC_ and VE_SYMP,_ and to indirectly estimate VE_INF_ using viral load, as each participant would contribute to the study endpoints. Human challenge studies have been widely used to better understand the natural history and treatment of various infections including malaria [67], influenza [68,69,70], RSV [71], and most recently, SARS-CoV-2 [72]. This approach could potentially be completed in fewer participants within 2–3 months, depending on the selected time between vaccination and viral challenge. While challenge studies are efficient, there are important ethical considerations regarding potential harm to study participants which must be weighed against the benefits of accruing important data more rapidly. Moreover, it will be uncertain whether results can be generalized to the wider population, particularly those in different age cohorts. Nevertheless, even crude estimates of VE_SUSC_, VE_SYMP_ and VE_INF_ could add critical knowledge to influence vaccine implementation policies.

Our approach has limitations. We exclude details pertaining to new circulating variants of concern other than B.1.1.7, particularly B.1.167.2, which now predominates globally. However, these concepts remain even more relevant in this context. VE_DIS_ has decreased against the delta variant [6,61], and it is vital to understand whether this represents a loss of efficacy against infection or merely against symptoms, and whether there is also a reduction in protection against secondary transmission.

The model reflects population conditions unique to King County Washington and is not equipped to make precise vaccine schedule assessments for different locations and is not meant as a predictive tool. Rather, we make the conclusion that VE_INF_ could theoretically provide substantial population-level benefits and provide a framework for the most rapid evaluation of this metric. We also do not consider all possible vaccine efficacies, including reduction but not elimination of symptoms, or reduction in severe disease. In addition, we lacked sufficient data to consider other important population subsets including gender, race, ethnicity, and immunosuppressed state.

Regarding our intra-host transmission modeling, we note that variables other than viral load may dictate transmission likelihood including duration and intensity of aerosol exposure. The relative infectiousness of the virus in asymptomatic people is another area of uncertainty in our model projections and this value might shift with emergence of new variants of concern. Our intra-host model was fit to early viral load data from the pandemic and has not been updated for new variants which may have higher viral loads [73,74,75]. Other models with different assumptions fit to separate sets of viral kinetic data well [25,76,77], and may provide slightly different results when considering inter-host transmission probabilities. Overall, our simulations are not intended as forecasts, but rather projections under different scenarios to allow qualitative conclusions on the role of various vaccine efficacy measures on SARS-CoV-2 incidence.

In conclusion, when observed high VE_DIS_ is predominately due to reduction in symptoms rather than absolute protection against infection, VE_INF_ will be important to measure, as it may determine the severity of subsequent waves of infections and deaths. Using peak viral load as a proxy measure in human challenge studies may be an efficient way to complement other clinical trial designs to assess VE_INF_.

## Figures and Tables

**Figure 1 viruses-13-01921-f001:**
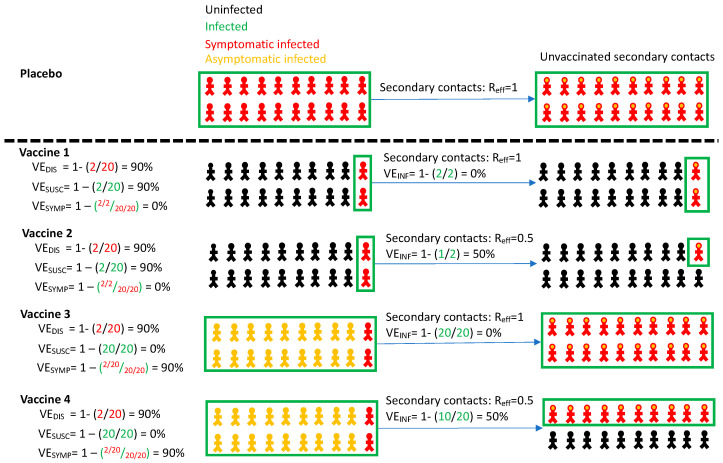
Vaccine efficacy definitions. Four vaccines with high efficacy against disease (VE_DIS_ = 90%) are demonstrated with different projected outcomes on vaccinated persons (**left**) and secondary contacts of infected person (**right**). Vaccines 1 and 2 mediate reduction of symptomatic infection by eliminating infection altogether, whereas vaccines 3 and 4 reduce symptoms among infected people. Vaccines 1 and 3 provide no reduction in secondary transmission risk. Vaccines 2 and 4 provide 50% reduction in secondary transmission risk. Definitions are in Table 1. All persons in the placebo arm are symptomatically infected for demonstration purposes only. Infected secondary contacts may be symptomatic or asymptomatic. Here, R_eff_ is the effective reproductive number representing number of secondary transmissions per infected person which we assume to be 1 in the absence of a vaccine.

**Figure 2 viruses-13-01921-f002:**
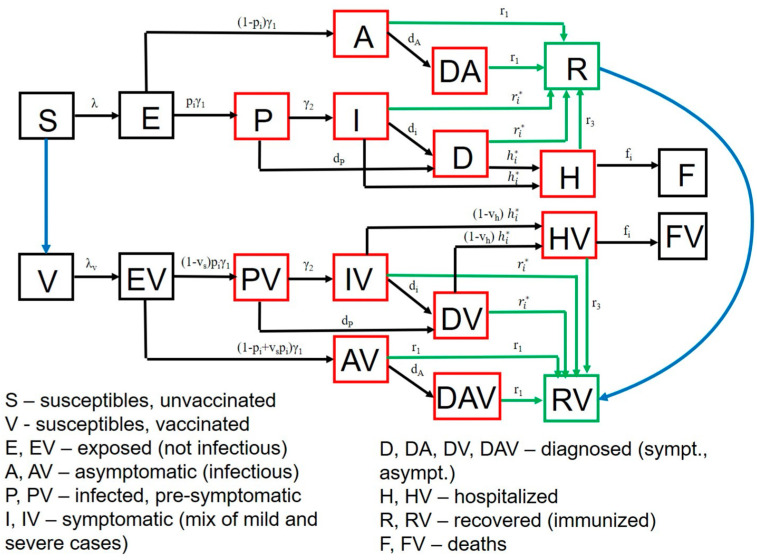
SARS-CoV-2 transmission model in King County, Washington. Model structure captures transition from susceptible (S) to exposed (E) to asymptomatic infection (A), or to pre-symptomatic (P) and then symptomatic infection (I) followed by recovery (R), hospitalization (H) or death (F). A similar potential pathway is also shown for a vaccinated cohort (V). Diagnosed (D) and diagnosed asymptomatic (DA) is an intermediate step for a proportion of people. Parallel versions of the model are run for variants with different infectivity.

**Figure 3 viruses-13-01921-f003:**
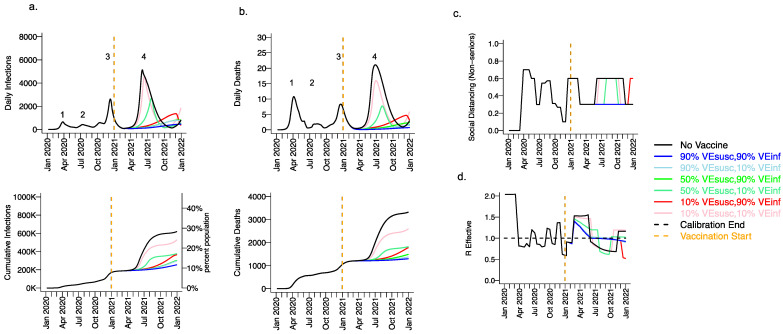
High VE_SUSC_ or high VE_INF_ can effectively limit infections and deaths with the B.1.1.7 variant. For unvaccinated (black lines) and each vaccine cohort (colored lines, legend), we project (**a**) infections, (**b**) deaths, as well as (**c**) social distancing relative to pre-pandemic levels and (**d**) the effective reproductive number. The first two columns (**a**,**b**) are organized by row: top = daily incidence, bottom = cumulative. Waves of infection are numbered 1–4. Six combinations of VE_SUSC_ and VE_INF_ are considered while VE_SYMP_ is fixed at 10%. High VE_SUSC_ (90%) simulations are blue and have similar outcomes to one another. Moderate VE_SUSC_ (50%) simulations are green. Low VE_SUSC_ (10%) simulations are red/pink. Dark lines are high VE_INF_ (90%). Light lines are low VE_INF_ (10%). The largest reduction in infections is associated with either high VE_SUSC_ or VE_INF_. 10,000 vaccines are given per day starting 1 January 2021 (orange line) until 90% are vaccinated in age groups other than children. Case threshold for reinstituting physical distancing to 0.6 is 350 per 100,000 and for relaxation is 100 per 100,000. 80% of vaccines are initially allocated to the elderly with the remaining 20% to middle-aged cohorts.

**Figure 4 viruses-13-01921-f004:**
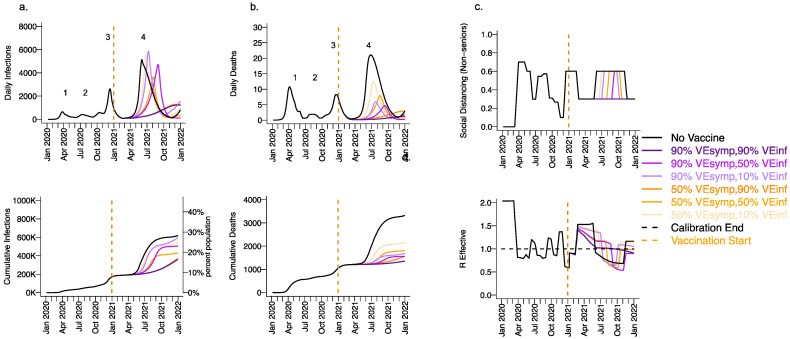
High VE_SYMP_ alone results in only partial reduction in infections and deaths with the B.1.1.7 variant. For unvaccinated (black lines) and each vaccine cohort (colored lines), we project (**a**) infections, (**b**) deaths, as well as (**c**) physical distancing relative to pre-pandemic levels and (**d**) the effective reproductive number. The first two columns (**a**,**b**) are organized by row: top = daily incidence and bottom = cumulative. Waves of infection are numbered 1-4. Six combinations of VE_SYMP_ and VE_INF_ are considered while VE_SUSC_ is fixed at low 10%. High VE_SYMP_ (90%) simulations are purple. Moderate VE_SYMP_ (50%) simulations are orange. Dark lines are high VE_INF_ (90%). Moderate darkness lines are medium VE_INF_ (50%). Light lines are low VE_INF_ (10%). The largest reduction in cases is associated with high VE_INF_. 10,000 vaccines are given per day starting 1 January 2021 (orange dashed vertical line) until 90% are vaccinated in age groups other than children. Case threshold for reinstituting physical distancing to 0.6 is 350 per 100,000 and for relaxation is 100 per 100,000. 80% of vaccines are initially allocated to the elderly.

**Figure 5 viruses-13-01921-f005:**
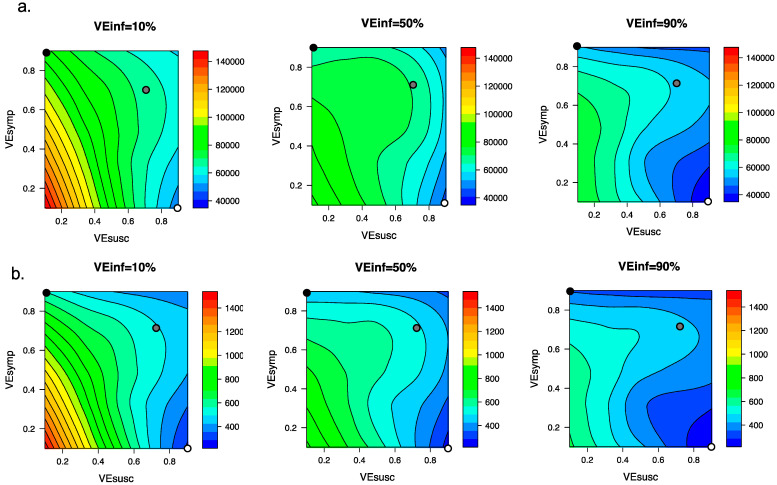
High VE_INF_ further reduces diagnosed cases and death only when VE_SUSC_ is low or moderate. Heat maps comparing contrasting vaccine scenarios. (**a**) Post-vaccine diagnosed cases (top row) which are approximately 25% of all infections and (**b**) post-vaccine deaths (bottom row) with different combinations of VE_SUSC_ and VE_SYMP_. In this simulation, there were 66558 diagnosed cases and 1054 deaths prior to vaccination and heat maps capture all outcomes beyond this point. The left column assumes VE_INF_ = 10%; middle column assumes VE_INF_ = 50%; right column assumes VE_INF_ = 90%. The dots are 3 scenarios compatible with results from the Pfizer and Moderna trials in which VE_DIS_ = 90% (black is VE_SYMP_ = 90%/VE_SUSC_ = 0%, grey is VE_SYMP_ = 70%/VE_SUSC_ = 70% and white is VE_SYMP_ = 0%/VE_SUSC_ = 90%). Increased VE_INF_ leads to a larger further reduction in cases when VE_DIS_ is mediated by high VE_SYMP_ than when it is mediated by high VE_SUSC_. In general, additional benefit of VE_INF_ is accrued when VE_SUSC_ is low, across a wide range of VE_SYMP_.

**Figure 6 viruses-13-01921-f006:**
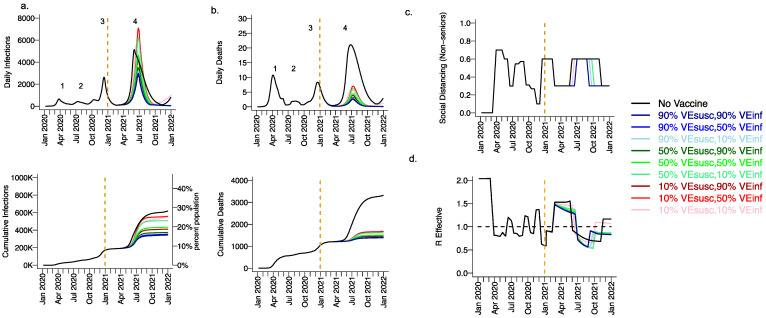
High VE_SUSC_ or high VE_INF_ limit the extent of a fourth wave at low vaccine roll out rates with the B.1.1.7 variant. For unvaccinated (black lines) and each vaccine cohort (colored lines, legend), we project (**a**) infections, (**b**) deaths, (**c**) social distancing relative to pre-pandemic levels and (**d**) the effective reproductive number. The first two columns (**a**,**b**) are organized by row: top = daily incidence, bottom = cumulative. Waves of infection are numbered 1-4. Six combinations of VE_SUSC_ and VE_INF_ are considered while VE_SYMP_ is fixed at 90% such that all vaccines would produce results consistent with those in the Pfizer and Moderna trials. High VE_SUSC_ (90%) simulations are blue. Moderate VE_SUSC_ (50%) simulations are green. Low VE_SUSC_ (10%) simulations are red/pink. Dark lines are high VE_INF_ (90%). Light lines are low VE_INF_ (10%). The largest reduction in cases is associated with either high VE_SUSC_ or VE_INF_. 5000 people are fully vaccinated per day starting 1 January 2021 (orange dotted vertical line) until 90% are vaccinated in age groups other than children. Case threshold for reinstituting physical distancing to 0.6 is 350 per 100,000 over two weeks and for relaxation is 100 per 100,000. 80% of vaccines are initially allocated to the elderly.

**Figure 7 viruses-13-01921-f007:**
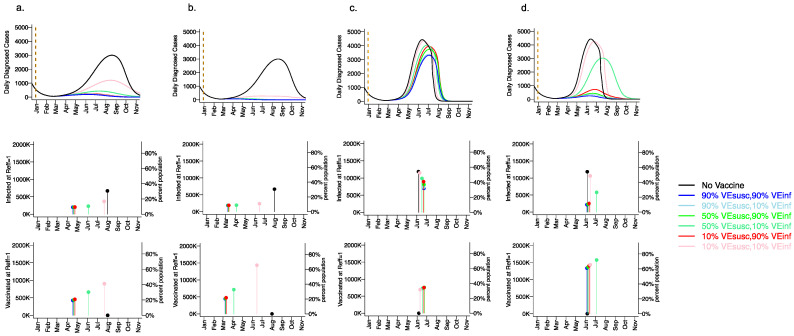
Rapid vaccine rollout rate and high VE_SUSC_ or VE_INF_ lower the number of infections prior Table 1. 1.7 variant. For unvaccinated (black lines) and each vaccine cohort (colored lines, legend), we project daily diagnosed cases during 2021 (top row); as well as timing of herd immunity threshold and number of infected at herd immunity threshold (middle row), and number of vaccinated at herd immunity threshold (bottom row) signified by dots and lines. Columns (**a**,**b**) assume the Wuhan variant while columns (**c**,**d**)assume the 55% more infectious B.1.1.7 variant with 55% greater infectivity leading to more rapid accrual of cases. In (**a**,**c**), 5000 people are fully vaccinated per day starting 1 January 2021 (orange dotted vertical line) until 90% are vaccinated in age groups other than children. In (**b**,**d**), 10,000 people are fully vaccinated per day starting 1 January 2021 (orange dotted vertical line) until 90% are vaccinated in age groups other than children. VE_SYMP_ is fixed at 90% in all simulations such that all vaccines would produce results consistent with those in the Pfizer and Moderna trials. High VE_SUSC_ (90%) simulations are blue. Moderate VE_SUSC_ (50%) simulations are green. Low VE_SUSC_ (10%) simulations are red/pink. Dark lines are high VE_INF_ (90%). Light lines are low VE_INF_ (10%). The largest reduction in cases is associated with either high VE_SUSC_ or VE_INF_. 80% of vaccines are initially allocated to the elderly. No reactive lockdown is assumed in these simulations such that herd immunity threshold is reached by virtue of cases and vaccinations.

**Table 1 viruses-13-01921-t001:** Vaccine efficacy definitions.

	Definition	Formula
Vaccine efficacy against symptomatic infection	Reduction in virologically confirmed symptomatic COVID-19 in vaccine versus placebo recipients	VE_DIS_ = 1 − (V_DIS_/P_DIS_)
Vaccine efficacy against all infection	Reduction in virologically confirmed asymptomatic or symptomatic SARS-CoV-2 infection in vaccine versus placebo recipients	VE_SUSC_ = 1 − (V_SUSC_/P_SUSC_)
Vaccine efficacy against symptoms given infection	Reduction in development of symptoms conditional on infection in vaccine versus placebo recipients	VE_SYMP_ = 1 − (V_DIS_/V_SUSC_)/(P_DIS_/P_SUSC_)
Vaccine efficacy against transmissability given infection	Reduction in number of secondary contacts infected by infected vaccine recipients versus number of secondary contacts infected by infected placebo recipients	VE_INF_ = 1 − (V_INF_/P_INF_)

V_DIS_ = % in the vaccine arm with virologically confirmed symptomatic COVID-19. P_DIS_ = % in the placebo arm with virologically confirmed symptomatic COVID-19. V_SUSC_ = % in the vaccine arm with virologically confirmed symptomatic or asymptomatic SARS-CoV-2 infection. P_SUSC_ = % in the placebo arm with virologically confirmed symptomatic or asymptomatic SARS-CoV-2 infection. V_INF_ = number of secondary contacts infected per infected vaccine recipients with virologically confirmed symptomatic or asymptomatic COVID-19. P_INF_ = number of secondary contacts infected per infected placebo recipients with virologically confirmed symptomatic or asymptomatic COVID-19.

## Data Availability

All model code is available: https://github.com/ashish2goyal/SARS_CoV_2_Super_Spreader_Event and https://github.com/FredHutch/COVID_modeling_sensitivity.

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
