# Peer review of "Mathematical Modeling of Vaccines That Prevent SARS-CoV-2 Transmission"

_viruses, 2021, doi:10.3390/v13101921_

Round 1

Reviewer 1 Report

The manuscript entitled « Mathematical modeling of vaccines that prevent SARS-CoV-2 transmission by lowering viral load” is an in silico study which aims at identifying the vaccines protocols that are most effective and the impact of simulated vaccination campaigns on lockdowns. The authors based their simulations on previous models developed for the King County Washington.

While the manuscript is overall well written, I have the following comments:

General comment:

  • I am not sure to understand the novelty of this study. The authors used previously existing models to deterministically simulate epidemics. Hence, most of the results that they present were expected from the model parametrization. In addition, most of the content of the result sections are in fact methodological precision and the actual result content is hence limited. Overall the structure of the manuscript should be entirely revised to help the reader understand this study.

Introduction:

  • The authors starts their introduction with the presentation of different definitions of vaccine efficacy. Some information should be specified or corrected. For instance, VESYMP is “defined by the authors as the reduction in the presence of symptoms […]”, however a reduction in symptom intensity is also crucial and should not be ignored.
  • Lines 55-58 : The authors state : “If a vaccine mediates VEDIS primarily through reduction in symptoms, the extent to which people, who convert from symptomatic to asymptomatic infection as a result of receiving the vaccine, can still transmit the virus, remains unknown.” Do the authors consider subjects who would have been symptomatic without symptoms but are asymptomatic with the vaccine? Or do they consider a reduction in symptom scores and a resolution of the clinical disease?
  • The authors focused their study on the King County, however it would be interesting for the reader to expose the public health policy at the level of United States and in other countries where the mitigation measures taken were different.
  • The tables and figures should be in the methods.

Methods:

  • The author should present the King County as well as tits demographic characteristics to the reader rather than provide references.
  • A graphical representation of the transmission model in the main text would be appreciated. Similarly, the within-host model should be presented in the main text.
  • It would be interesting to do sensitivity analyses to describe the impact of the fixed parameters (I50, ω…) on the final outputs.
  • Overall section 2.6 should be entirely rewritten since it is not clear what the authors exactly do.
  • Line 145 : the status “dead” is missing
  • Line 182 : the authors should not ask the reader to go to the result section to have details about the methods. This reveals the lack of the structure of the manuscript
  • Line 199: what are M1 and M2? Please define.
  • Line 230: n1=N/5. Is N the N presented previously and equal 30? Please specify
  • Line 231 : “If more than half of the animals allocated to V1 are infected […]”: what is the complementary condition? It is not specified in the manuscript.

Results:

  • Please delete element of the Figures’ captions included in the main text.
  • Paragraph 3.1 should be in the introduction.
  • Paragraph 3.2 is redundant: vaccine efficacy definitions were already presented in the introduction and in the method.
  • Lines 297 – 319 are redundant: this was already presented in the methods.
  • Figure 2: I could not read the caption. However, if not done please specify the model and parameters. What are the black star and the orange square?
  • Lines 320 – 331 : the
  • Lines 341 – 360 are redundant: this was already presented in the methods. In addition, references are need for the lines 348 to 353.
  • Lines 364- 365 should be detailed in the methods.
  • Lines 370-379 are redundant: this was already presented in the methods.
  • Paragraph 3.10 should be in the methods
  • Line 505: Hill functions are designed to produced sigmoidal
  • Lines 513 -518 should be in the discussion
  • “We performed mock experiments» should be rephrased to clarify that these were in silico experiments.
  • Paragraph 3.12 : The design of experimental infection described here would not mimic a naturally acquired infection. Indeed, with experimental infection by inoculation, the inoculated dose is usually much larger than the amount of virus subjects infected by contact are exposed to.

Discussion:

The heterogeneity of the population, particularly considering the immune response (immunosenescence, immunodepressed individuals…) and clinical expression of the disease (age-related), as not been considered. These two aspects are crucial to predict the impact of the epidemics and the impact of the vaccination campaigns.

The authors assume that infectiousness is related to viral load only. However, it could be hypothesized that symptoms increasing the spread of infectious particles (sneezing, coughing) could enhance the infectiousness. The infectiousness would therefore result from the amount of shed virus and respiratory symptom scores. Likewise, the contamination of the environment and the role of fomites in transmission have not been explored.

The ethical limitations associated with human experimental challenged should also be exposed.

Globally, the discussion lacks many references.

Minor comments :

  • Several typos throughout the manuscript

Author Response

Reviewer 2

 Comments and Suggestions for Authors

The manuscript entitled « Mathematical modeling of vaccines that prevent SARS-CoV-2 transmission by lowering viral load” is an in silico study which aims at identifying the vaccines protocols that are most effective and the impact of simulated vaccination campaigns on lockdowns. The authors based their simulations on previous models developed for the King County Washington.

While the manuscript is overall well written, I have the following comments:

General comment:

I am not sure to understand the novelty of this study. The authors used previously existing models to deterministically simulate epidemics. Hence, most of the results that they present were expected from the model parametrization. In addition, most of the content of the result sections are in fact methodological precision and the actual result content is hence limited. Overall the structure of the manuscript should be entirely revised to help the reader understand this study.

We made efforts to restructure the article according to suggestions below. Specifically, we focus on generalizable features of our model to point out that vaccines with high VE_Susc or VE_inf allow a more rapid path to achieving herd immunity thresholds with fewer cases overall.

Introduction:

The authors starts their introduction with the presentation of different definitions of vaccine efficacy. Some information should be specified or corrected. For instance, VESYMP is “defined by the authors as the reduction in the presence of symptoms [...]”, however a reduction in symptom intensity is also crucial and should not be ignored.

We agree that a vaccine which reduces symptoms but does eliminate them would be of some value. However, we are most concerned with reduction in cases, hospitalizations, and deaths. We now acknowledge that reduction in mild symptomatic cases is not captured by our approach and this is a limitation.

Lines 55-58 : The authors state : “If a vaccine mediates VEDIS primarily through reduction in symptoms, the extent to which people, who convert from symptomatic to asymptomatic infection as a result of receiving the vaccine, can still transmit the virus, remains unknown.” Do the authors consider subjects who would have been symptomatic without symptoms but are asymptomatic with the vaccine? Or do they consider a reduction in symptom scores and a resolution of the clinical disease?

The model definition of VEsymp assumes that the vaccine converts its recipient from symptomatic to completely asymptomatic with a subsequent 56% reduction in infectivity. This was stated in the last version and is now re-emphasized more clearly in the resubmitted version.

The authors focused their study on the King County, however it would be interesting for the reader to expose the public health policy at the level of United States and in other countries where the mitigation measures taken were different.

It is beyond the scope of the current paper to simulate multiple epidemics precisely. However, we think the idea to make the model more generalizable is a good one. To this end, we have conducted new simulations in which we consider how the following variables which might impact time to the herd immunity threshold: vaccine efficacy, and vaccination rate. Future versions of the model will also assess how degree of ongoing physical distancing and baseline seroprevalence prior to vaccination impact the timing of sustainment of the herd immunity threshold. Please see Figure 7 for these results.

The tables and figures should be in the methods.

Typically, most figures and tables are included in the results sections of papers. In this case, we moved Figure 1 to the introduction and Figure 2 to the methods to clarify our message.

Methods:

The author should present the King County as well as tits demographic characteristics to the reader rather than provide references.

We agree and have now done so in the Supplement tables.

A graphical representation of the transmission model in the main text would be appreciated. Similarly, the within-host model should be presented in the main text.

We have moved the epi model schematics to the main text in Figure 2. The intra-host model is now only included in the supplement. We reference a published schematic of the model.

It would be interesting to do sensitivity analyses to describe the impact of the fixed parameters (I50, ω...) on the final outputs.

In the revised version, we focus less attention on the viral dynamics model which has already been published and analyzed elsewhere so chose not to perform sensitivity analyses,

Overall section 2.6 should be entirely rewritten since it is not clear what the authors exactly do.

We have now eliminated this section for brevity

Line 145 : the status “dead” is missing

We have added this.

Line 182 : the authors should not ask the reader to go to the result section to have details about the methods. This reveals the lack of the structure of the manuscript

Efficacy profiles are now described in the methods.

Line 199: what are M1 and M2? Please define.

These immune cell populations are a feature of our published model but are not included in the more simplified version of the intra-host model described in this paper.

Line 230: n1=N/5. Is N the N presented previously and equal 30? Please specify

This section is now eliminated to allow greater focus in the manuscript.

Line 231 : “If more than half of the animals allocated to V1 are infected [...]”: what is the complementary condition? It is not specified in the manuscript.

This section is now eliminated to allow greater focus in the manuscript.

Results:

Please delete element of the Figures’ captions included in the main text.

This is for ease of reading and is kept in

Paragraph 3.1 should be in the introduction.

We now have moved much of this content to the introduction and methods.

Paragraph 3.2 is redundant: vaccine efficacy definitions were already presented in the introduction and in the method.

These are now detailed only in the methods

Lines 297 – 319 are redundant: this was already presented in the methods.

These are now detailed only in the methods

Figure 2: I could not read the caption. However, if not done please specify the model and parameters. What are the black star and the orange square?

We apologize for the small caption and have made the requested changes. The star and square are eliminated in favor of vertical lines which are labelled.

Lines 320 – 331 : the

Lines 341 – 360 are redundant: this was already presented in the methods. In addition, references are need for the lines 348 to 353.

Lines 364- 365 should be detailed in the methods.

Lines 370-379 are redundant: this was already presented in the methods.

Paragraph 3.10 should be in the methods

These sections are now eliminated from the results

Line 505: Hill functions are designed to produced sigmoidal

Lines 513 -518 should be in the discussion

This section is now not included

“We performed mock experiments» should be rephrased to clarify that these were in silico experiments.

Paragraph 3.12 : The design of experimental infection described here would not mimic a naturally acquired infection. Indeed, with experimental infection by inoculation, the inoculated dose is usually much larger than the amount of virus subjects infected by contact are exposed to.

We agree that there are some artificial components of challenge studies. However, extensive experience with RSV and influenza challenge models reveals that challenge studies provide a realistic approximation of infection. Nevertheless, we now only discuss challenge studies in the discussion as a point of take off for future research.

Discussion:

The heterogeneity of the population, particularly considering the immune response (immunosenescence, immunodepressed individuals...) and clinical expression of the disease (age-related), as not been considered. These two aspects are crucial to predict the impact of the epidemics and the impact of the vaccination campaigns.

The model has been calibrated to capture the overall effects of these variables at the population level. It is far beyond the scope of this modeling exercise, and beyond our scientific question, to address these variables in our model. We therefore list this as a limitation of our approach. We do have age stratified mortality and hospitalization rates which are the most critical determinants for this virus.

The authors assume that infectiousness is related to viral load only. However, it could be hypothesized that symptoms increasing the spread of infectious particles (sneezing, coughing) could enhance the infectiousness. The infectiousness would therefore result from the amount of shed virus and respiratory symptom scores. Likewise, the contamination of the environment and the role of fomites in transmission have not been explored.

We cited numerous sources indicating that viral load is a critical determinant of transmission risk but concur that other variables pertaining to aerosolization risk could be important and discuss this as a limitation in the revision.

The ethical limitations associated with human experimental challenged should also be exposed.

These are discussed in more detail in the revised version.

Globally, the discussion lacks many references. Minor comments :
Several typos throughout the manuscript

Reviewer 2 Report

Two files should be attached: a word file and a pdf of the manuscript with a multitude of comments.

Author Response

Reviewer 3

 Review of “Mathematical modeling of vaccines that prevent SARS-CoV-2 transmission by lowering viral load”

This paper starts with a good delineation of vaccine effect measures and illustrates the nature of these so that readers of various levels of sophistication can understand them. Unfortunately, the terms used for the different measures discussed are non-standard and confusing. Those terms even seem to have confused the authors as they have to different measure expressing vaccine efficacy against infection.

We apologize for our miswording in Table 1. We do think our definitions for vaccine efficacy are standard as described here.

Still, this paper lays out what measures can and cannot be estimated from data collected in vaccine trials. It argues that vaccine efficacy against infection is a key vaccine effect measure predicting future effects of vaccination. It also argues that vaccine effect is not measurable from current vaccine trials and would take intense trial testing of asymptomatic individuals at a high frequency to measure the occurrence of asymptomatic infection. Finally, it argues that that measuring vaccine levels in intentionally exposed volunteers would provide key data that would allow for inferring vaccine efficacy against infection. Such volunteer experiments would have controversial ethics and likely would not be undertaken just as volunteer exposure for initial vaccine efficacy assessment was never undertaken. But the argument for such volunteer experiments deserves to be laid out. If the authors wanted to make an argument for these human challenge studies, they would have to compare the costs and consequences of alternative approaches like intense follow up of a large number of vaccinated and unvaccinated individuals.

This point is very well-taken and we do not feel as if our paper is equipped to argue in favor or against human challenge studies. Rather, we only wish to make the point that such studies could be performed in a relatively small number of participants. In the revised paper, we specify that human challenge studies are only one option to arrive at this metric and limit this writing to the discussion. We highlight the various ethical and pragmatic challenges with this approach and that these would need to be balanced relative to other study approaches.

Despite these problems, the goals of this paper are excellent and laudable. The paper would be useful even if the final argument about volunteer human challenge studies was not part of it. But that final argument is important for making decisions about what data should be pursued. Unfortunately, that final argument is exceedingly muffled and unclear and too summarily presented.

Please see our statement above. We no longer include this section as an argument but rather to demonstrate that significant virological differences could be demonstrated with a relatively small sample size. We leave it to the discretion of the reader what approach would be most appropriate for defining various vaccine efficacies as many of the variables guiding these decisions are beyond the scope of our work.

Our primary point is that identifying our 3 defined vaccine efficacies should be a major priority as they will allow dramatically different outcomes at a given vaccination rate and viral variant infectivity.

A specific problem is that the model used for within host viral dynamics is one appropriate for cellular immunity. But complete elimination of infectiousness requires a humoral response. For some infections a cellular immune response may prevent infectiousness. But not for SARS-CoV-2. The incubation period is just too short.

We respectfully somewhat disagree with this statement. We agree that humoral immunity is likely to play a key role in preventing infection. Yet, a vaccine that reduces peak viral load could work via either acquired immune mechanism including pre-existing humoral immunity or a tissue-resident T cell response. The mechanism for reduction in peak SARS-CoV-2 viral load due to vaccination is currently unknown. As described in our Science Advances paper (PMID: 33097472), we are also able to model equivalent viral dynamics whether assuming humoral or cell-mediated immunity. For the sake of viral transmission modeling, the priority is generating realistic viral trajectories rather than explaining the true mechanism of immunity. We explain this in the revised manuscript.

The article has been on medRxiv for some time with a different format but most of the present content. If it was reviewed elsewhere, those reviews should be made public. The thesis of this paper is too important for it to be allowed to drift about without public comment. Time is passing and the pandemic control through vaccination could reach a critical stage to which the world should be prepared to respond. Thus, it would help to make the discussion of this paper open now. The problem is the paper in its current form is too difficult to understand. Hopefully others will be motivated to understand the paper. But all epidemiologist and infection transmission system modelers we are too busy with this pandemic so that only those formally charged with making a review will take the time to view it critically.

We really appreciate this comment. Frankly, our group has struggled with the balance of writing manuscripts in a typical academic fashion while also releasing them early for maximal public health benefit (all while keeping up with our non-COVID research). To this end, this manuscript has received some attention on medRxiv but this represents our first formal review.

Our opinion is that the immediate potential public health impact of the paper may have passed, at least in King County where vaccination acceptance is extremely high. However, the message is still of high importance particularly given the emergence of highly infectious new global variants. While the true values for VE_susc, VE_inf and VE_symp of licensed vaccines are not yet precisely defined, there appears to a be a consensus based on observational data from around the globe that the vaccine has moderate to high VE_susc and VE_inf. Moreover, the vaccine roll out has occurred extremely rapidly and masking mandates and restrictions on physical distancing are being lifted across the country. It is therefore possible that vaccine efficacy characteristics could determine the extent of a fifth wave in the fall and winter.

We therefore prefer to frame the manuscript in its present form as an academic contribution which makes the point that increases in VE_susc (or Ve_inf in the case where observed trial efficacy is driven mostly by VE_symp) will lower the total number of infections and to a lesser extent hospitalizations and deaths and also decrease the number requiring vaccination prior to reaching the herd immunity threshold. We also point out that the limited 4th wave that has occurred in King County are compatible with vaccines that have high values for VE_susc.

While our model is not capable of estimating the herd immunity threshold in other local epidemics across the globe, it can at least make the point that assuming that high protection against secondary infection is maintained against evolving variants, a lower percentage of people must be vaccinated to achieve the herd immunity threshold.

I have made a first stab at this. I have made very extensive comments on the pdf of the manuscript.

The paper can be saved. My extensive notes on the parts of the paper that I have highlighted could be followed to make this paper more comprehensible and to give it a higher impact. The following are a few broad steps from those notes. Other of my notes might be even more important.

Thank you for this effort. We have incorporated many of the suggested changes in the updated version.

  • Cut the paper drastically.

We agree and cut several sections of the paper and focus primarily on the message of how vaccine efficacies impact the time to the herd immunity threshold. In a much briefer fashion, we outline possible methods to conduct studies using viral load to estimate the three different vaccine efficacies described herein. This is all limited to the discussion.

  • Focus on one simple theoretical or policy choice issue in a way that makes the objectives clear. After accomplishing that, other purposes can be discussed. Currently there are a great variety of confusing objectives. Predicting the fourth wave seems not to be as important an objective as getting good estimates of the vaccine effect measures.

We now focus on the herd immunity threshold as this could provide policy makers with a benchmark for timing and rapidity at which it is safe to relax social distancing measures.

  • Relate the vaccine effect measures presented here to those that stand out in the literature. Specifically the measures presented here should be compared to their nearest measures in a standard text like Design and Analysis of Vaccine Studies (Statistics for Biology and Health) by Halloran et al.

We tried to more clearly define the vaccine efficacies in the revised version. These were obtained directly from Dr. Halloran’s original paper on the subject.

  • The different models used need to have the same symbols for parameters that are the same. And they all need to be presented in a form that allows the reader to see how they relate to each other. One reason the argument is so muffled is because references are made to other models where it is impossible to see how the models relate to each other. The worst situation is where reference is made to the Science Advances paper that really does not address the issue in this paper. It is difficult to see what in this paper is essential. For that reason, at least a couple of the fitted curves in Figure 1 of that paper should be presented and how that fitting will contribute to the overall objectives of this paper needs to be presented.

We have now put all results from the intra-host model into the supplement. We make substantial effort to describe that this model is entirely separate from the epidemiologic model of King County and that we have published on this model on 3 separata occasions. All viral load modeling is limited to the supplement to focus mostly on the epidemiology.

  • The data that needs to be collected from humans in the human challenge studies needs to be clearly defined and exactly how that data should be used must be clearly illustrated. To argue for human challenge studies, one must also outline the observational data or experimental data in other species that would be needed to make the crucial estimates. In other

We de-emphasize any sample size analysis for human challenge studies in the revision and rather bring this up in the discussion as a topic for future research.

  • The preliminary description of scenarios in section 3.4 is not very helpful. Just jump right into the various scenarios. And just present these as theoretical cases. There is no way in the presence of the unknowns that they can be used to guide policy. The point of the paper should be how important the data you propose to gather in human challenge studies is. It will be important not only for predicting the fourth wave but for designing control after the fourth wave as well. For each scenario you should comment on this importance if that in fact is a main point of the paper. (You need to decide on a single main point of the paper and stick to it or the paper will remain in its current incomprehensible state.)

We again emphasize that our paper does not provide specific enough information to guide policy but rather to make the scientific points outlined above. We emphasize throughout that we are making projections rather than forecasts.

Reviewer 3 Report

Summary:

In their manuscript, Swan et al. have pointed out that vaccine efficiency as currently estimated in vaccine trails (VEdis) does not provide enough information to model and predict the effects of vaccination campaigns. They show, based on a mathematical model specific for King County (Washington), how different vaccine effects (VEsymp, VEsusc and VEinf) influence the dynamics of the epidemic even if VEdis values are similar. Moreover, they propose an experimental set-up to estimate VEinf. I think that this is a valuable contribution to the current literature.

I do have some major and minor comments on the manuscript, which I have summarized below.

Major comments:

At several locations in the manuscript, the authors emphasize the importance of VEsusc, VEsymp and VEinf on the fraction of individuals that need to be vaccinated to obtain herd immunity in the population. Would it be possible to provide intuition on how these factors influence the herd immunity? Or would it be even be possible to derive this quantity?

More specifically, I would also find it interesting to see how the model parameters are associated with this fraction of individuals that need to be vaccinated to obtain herd immunity. How does the reproduction number of the disease interact with the characteristics of the vaccine to determine how many individuals need to get vaccinated to obtain herd immunity?

Also, how sensitive are the model results for the other model parameters? This would help to understand to which extend the results on the impact of VEinf, VEsusc and VEsymp on the dynamics of the epidemic are generalizable to other diseases.

Minor comments:

Table 1 has the same description in the left column for VEinf and VEsusc, I find this confusing. I think VEinf should be defined as the efficiency against infectiousness instead of infection.

Supplementary table 1 could be extended with a reference for the fixed values. Even though the main text mentions some citations for the calibration of the model parameters, it is not clear to me which parameter is taken from which reference.

Why did the authors decide to estimate most of the parameters, after an initial period, per month? Is there for instance any evidence that the case fatality rate is expected to differ per month?

Could the y-axis of supplementary Figure 6 be shown on a logarithmic scale?

Lastly, I found some typos in the manuscript:

Line 147 ‘Poarameter’

Line 222 ‘does’

Line 393 ‘VEdIS’

Line 513: ‘the’ is missing

Author Response

Reviewer 1:

Summary:

In their manuscript, Swan et al. have pointed out that vaccine efficiency as currently estimated in vaccine trails (VEdis) does not provide enough information to model and predict the effects of vaccination campaigns. They show, based on a mathematical model specific for King County (Washington), how different vaccine effects (VEsymp, VEsusc and VEinf) influence the dynamics of the epidemic even if VEdis values are similar. Moreover, they propose an experimental set-up to estimate VEinf. I think that this is a valuable contribution to the current literature.

Thank you for this positive feedback.

I do have some major and minor comments on the manuscript, which I have summarized below.

Major comments:

At several locations in the manuscript, the authors emphasize the importance of VEsusc, VEsymp and VEinf on the fraction of individuals that need to be vaccinated to obtain herd immunity in the population. Would it be possible to provide intuition on how these factors influence the herd immunity? Or would it be even be possible to derive this quantity?

Thank you for pointing this out. We agree that an important aspect of the work is that we identify that vaccines which have either high VEsusc or high VEinf mean that far fewer vaccinations are needed to reach a herd immunity threshold where Reff<1. In the revised version of the manuscript (Figure 7), we explore this concept further by including scenarios with slow and rapid vaccine roll out, and lower and higher infectivity variants. We identify that rapid vaccination with vaccines having higher VEsusc or VEinf allow fewer infections prior to attainment of the herd immunity threshold.

Based on our simulations, we do not think it is possible to identify a precise value capturing how these three metrics impact the herd immunity threshold for vaccination because this threshold is dependent on characteristics of local epidemics aside from the vaccine including use of non-pharmaceutical interventions), the infectivity of the dominant variant and baseline immunity due to past waves of infection. We can only make the broad generalization that vaccines with high VEsusc or VEinf lower the number of people requiring vaccination to achieve the herd immunity threshold.

Please see Figure 7 for details of this analysis.

More specifically, I would also find it interesting to see how the model parameters are associated with this fraction of individuals that need to be vaccinated to obtain herd immunity. How does the reproduction number of the disease interact with the characteristics of the vaccine to determine how many individuals need to get vaccinated to obtain herd immunity?

We agree that this is interesting. When we re-simulated the model with the 55% more infectious B.1.1.7 variant, we identified that a considerably higher percentage of people need to be vaccinated to reach the herd immunity threshold given an equivalent vaccine efficacy profile and vaccination rate.

Notably at higher vaccination rates, differences in time to reaching the vaccination threshold for her immunity are relatively diminished but higher vaccination rate leads to fewer infections when the threshold is surpassed. These points are now all emphasized in the text of the results in the revised version.

Of note in the revised paper, for figure 7, we assume no reactive lockdowns to focus purely on the impact of vaccination rather than non-pharmaceutical interventions. This is largely consistent with the trajectory of the current fourth wave during which society is generally opening.

Also, how sensitive are the model results for the other model parameters? This would help to understand to which extend the results on the impact of VEinf, VEsusc and VEsymp on the dynamics of the epidemic are generalizable to other diseases.

We have now published a sensitivity analysis of the King County model elsewhere (PMID: 33899042). Here we focus on what impact changes in vaccination roll out rate, vaccine efficacy and R_eff of novel variants have on time and number of vaccinations prior to reaching the herd immunity threshold.

Minor comments:

Table 1 has the same description in the left column for VEinf and VEsusc, I find this confusing. I think VEinf should be defined as the efficiency against infectiousness instead of infection.

We are truly sorry for this oversight and have corrected the table.

Supplementary table 1 could be extended with a reference for the fixed values. Even though the main text mentions some citations for the calibration of the model parameters, it is not clear to me which parameter is taken from which reference.

Thank you. We have updated the table.

Why did the authors decide to estimate most of the parameters, after an initial period, per month? Is there for instance any evidence that the case fatality rate is expected to differ per month?

This decision was made based on the possibility that several factors could alter case fatality rate over time including a healthier segment of the population getting infected, advances in medical care and lower overall inoculum doses due to masking. This is explained in the revision.

Could the y-axis of supplementary Figure 6 be shown on a logarithmic scale?

We amended the y-axis but found that the non-log converted axes more clearly conveyed our message

Lastly, I found some typos in the manuscript: Line 147 ‘Poarameter’
Line 222 ‘does’
Line 393 ‘VEdIS’

Line 513: ‘the’ is missing

Thank you. These are all corrected.

Round 2

Reviewer 1 Report

The manuscript has greatly improved and is much easier to read. The authors made extensive changes in their manuscript and this is much appreciated. However I still have some comments about the manuscript.
The main point I want to raise is the discussion about the human challenges (also mentioned by reviewer #3). I agree with his/her comments and I think that the discussion is not mentioning enough that these studies are very rare (to my knowledge only one human challenge with influenza has been done since the 90’s.) and that it is highly improbable that ethic committee would allow such experiments. The authors added a piece of sentence to nuance their paragraph, but this does not seem sufficient. In addition, the fact that experimental challenge are representative or not of naturally acquired infections should be at least discussed and supported by references (this was mostly explored in animal challenges).

I also still find that the manuscript lack of references to works performed by other teams. For instance, about the viral kinetics, the work from Néant et al. PNAS 2021 is not cited. This study could be at least discussed when the authors mentioned that the date of infection is not known, since this team suggested a method to back-calculate this parameter. Similarly, for the vaccine efficacy clinical trials, the Baden et al. and Polack et al. published in NEJM are cited, but additional studies were performed in general population since then. For instance, the paper by Haas et al. Lancet 2021 in which efficacy of the Pfizer-BioNTech vaccine has been studied in Israel.

Throughout the manuscript the authors speak about infectivity when I believe it should be infectiousness in some occasions. In deed, Infectivity expresses the ability of the disease agent to enter, survive and multiply in the host; infectiousness indicates the relative ease with which a disease is transmitted to other hosts (according to UCLA Fielding school of Public Health).

Lines 260-267 : could you please explain how the fixed values for omega and m were chosen? These are important hypotheses that should be explained and discussed.

The authors made substantial efforts to reorganize the manuscript and to delete redundant information that was presented in both methods and results. Some more paragraphs in the results are not needed or redundant : First paragraph of section 3.1 repeats what has been presented in the methods. The authors could instead describe the figure S1 (quality of adjustment…). Idem for lines 381-384 and 395-404. Similarly, section 3.4 is an interpretation of the results shown in Figs 3 & 4 and should rather be discussed later in the discussion. Instead, the authors could present the results shown in these figures (height and delay of the peak of infection & death …).

Finally, section 3.8 mostly relates results from previous works and not results from this study and should be deleted or moved to the discussion.

Minor comments:

Line 59: “convert from symptomatic to asymptomatic infection”: I previously raised this point, but I was probably not clear enough. As stated in the sentence, this suggest that individual that are symptomatic would see their symptoms resolve following a vaccination, and therefore this means that vaccination would act as a curative treatment. Where in fact the authors mean that individual that would have been symptomatic if they were not vaccinated, would be asymptomatic if they were vaccinated.

Table 1: The definitions of the different vaccine efficacies are well explained in the introduction and the additional details presented in this table are related to what is exposed in the methods. I find that it disturb the reading of the introduction and oblige the reader to go back to the introduction when this information would be valuable. The table itself is good but not well positioned in the manuscript.

Names of the vaccine: To avoid any confusion for the reader and in case industrials would develop new anti-Covid19 vaccines, the registration names of the different vaccines should be mentioned at least once (for instance BNT162b2 for the Pfizer-BioNTech vaccine, mRNA-1273QS for the Moderna vaccine …).

Line 146 : “We used an epidemiologic mathematical model of COVID-19” please specify that this is a transmission model.

Lines 166-167: the authors reformulated the names of their model’s compartiments, which is really appreciated. However these names can be confusing. For instance the compartments “symptomatic infected” and “diagnosed symptomatic” should rather be “undiagnosed symptomatic” and “diagnosed symptomatic” since in both cases the subjects are considered infected.

Lines 170-172 : “We assumed that 20% of infections are asymptomatic and that asymptomatic people are as infectious as symptomatic individuals but missing the highly infectious pre-symptomatic phase”. This is an important hypothesis and it should be discussed.

Throughout the manuscript: sometimes “within-host”, sometimes “intra-host” : please homogenize

Also please give exact result and not approximation as symbolized by “~”

I also persist about adding legend elements in the main text: This is really disturbing the reading. If the reader needs the information while looking at the figure, then everything is already explained in the legend (the legends and captions are very well detailed). It is hence in addition redundant.

Stating that the simulations represent variant B.1.1.7. seems too precise on the other hand. Since the difference between the “historical” and “B.1.1.7.” in the simulations result solely from a difference in infectivity, the authors should rather explain that they simulate “a variant” similar to the B.1.1.7. in terms of infectivity. In addition, the authors could explain that this approach would be easily applicable to other variants with other characteristics.

Line 414 : “With a vaccination rate of 5000 per day assuming the baseline variant, the number of 414 daily diagnosed cases was profoundly diminished…”. Could you please quantify “profoundly diminished”, please?

In addition, provided the population size of King County, would  help the reader to better apprehend what the vaccination rates represent.

Line 485 : “estimate” should be replaced by “compute” for instance since this result was calculated from the “estimated reduction in number of infection” and not “estimated” itself. I am also not sure that this is an estimation since it results from simulation…

Line 513 : “… this challenge remains relevant as new viral variants continue to emerge across the globe and these variants may result in lower vaccine efficacy.” Please provide a references

In the supplementary materials: please clearly state what M1 and M2 are. I thank you for explaining me what they were in your answer and I believe other readers would also be interested.

Figure S4: please increase font size.

Author Response

The main point I want to raise is the discussion about the human challenges (also mentioned by reviewer #3). I agree with his/her comments and I think that the discussion is not mentioning enough that these studies are very rare (to my knowledge only one human challenge with influenza has been done since the 90’s.) and that it is highly improbable that ethic committee would allow such experiments. The authors added a piece of sentence to nuance their paragraph, but this does not seem sufficient. In addition, the fact that experimental challenge are representative or not of naturally acquired infections should be at least discussed and supported by references (this was mostly explored in animal challenges).

  • Human challenge studies have been extensively performed for various human pathogens. We have now added language citing the full extent of human challenge studies that have occurred including pathogens such as RSV, SARS-CoV-2, influenza, and malaria.

I also still find that the manuscript lack of references to works performed by other teams. For instance, about the viral kinetics, the work from Néant et al. PNAS 2021 is not cited. This study could be at least discussed when the authors mentioned that the date of infection is not known, since this team suggested a method to back-calculate this parameter. Similarly, for the vaccine efficacy clinical trials, the Baden et al. and Polack et al. published in NEJM are cited, but additional studies were performed in general population since then. For instance, the paper by Haas et al. Lancet 2021 in which efficacy of the Pfizer-BioNTech vaccine has been studied in Israel.

  • We agree that the lack of inclusion of the Neant study is an oversight for which we apologize. We have updated the paper with this and other references of more recent viral kinetics studies.

Throughout the manuscript the authors speak about infectivity when I believe it should be infectiousness in some occasions. In deed, Infectivity expresses the ability of the disease agent to enter, survive and multiply in the host; infectiousness indicates the relative ease with which a disease is transmitted to other hosts (according to UCLA Fielding school of Public Health).

  • We agree that infectiousness is the better term and have changed this throughout

Lines 260-267 : could you please explain how the fixed values for omega and m were chosen? These are important hypotheses that should be explained and discussed.

  • Omega and m values were selected based on literature review and citations. We have now updated this section to cite these values.

The authors made substantial efforts to reorganize the manuscript and to delete redundant information that was presented in both methods and results. Some more paragraphs in the results are not needed or redundant : First paragraph of section 3.1 repeats what has been presented in the methods. The authors could instead describe the figure S1 (quality of adjustment…). Idem for lines 381-384 and 395-404. Similarly, section 3.4 is an interpretation of the results shown in Figs 3 & 4 and should rather be discussed later in the discussion. Instead, the authors could present the results shown in these figures (height and delay of the peak of infection & death …).

  • We appreciate the positive feedback. We have taken further effort to reduce overlap between the methods and results as suggested by the reviewer, though we do believe section 3.1 is showing results rather than methods and leave this section in an abridged format. In other cases, we made minimal changes because we felt it was important to highlight assumptions made that were specific to the simulations shown in Figure 7. We moved section 3.4 to the discussion and agree that this improves the flow of the paper.

Finally, section 3.8 mostly relates results from previous works and not results from this study and should be deleted or moved to the discussion.

  • We have significantly shortened this section and moved certain language to the methods.

Minor comments:

Line 59: “convert from symptomatic to asymptomatic infection”: I previously raised this point, but I was probably not clear enough. As stated in the sentence, this suggest that individual that are symptomatic would see their symptoms resolve following a vaccination, and therefore this means that vaccination would act as a curative treatment. Where in fact the authors mean that individual that would have been symptomatic if they were not vaccinated, would be asymptomatic if they were vaccinated.

  • This is correct and we have reworded this sentence for clarity.

Table 1: The definitions of the different vaccine efficacies are well explained in the introduction and the additional details presented in this table are related to what is exposed in the methods. I find that it disturb the reading of the introduction and oblige the reader to go back to the introduction when this information would be valuable. The table itself is good but not well positioned in the manuscript.

  • We have now shifted the Table to the Methods

Names of the vaccine: To avoid any confusion for the reader and in case industrials would develop new anti-Covid19 vaccines, the registration names of the different vaccines should be mentioned at least once (for instance BNT162b2 for the Pfizer-BioNTech vaccine, mRNA-1273QS for the Moderna vaccine …).

  • Thank you. We have made this change

Line 146 : “We used an epidemiologic mathematical model of COVID-19” please specify that this is a transmission model.

  • Thank you. We have changed this section accordingly

Lines 166-167: the authors reformulated the names of their model’s compartments, which is really appreciated. However these names can be confusing. For instance the compartments “symptomatic infected” and “diagnosed symptomatic” should rather be “undiagnosed symptomatic” and “diagnosed symptomatic” since in both cases the subjects are considered infected.

  • We have changed the compartment names for clarit.

Lines 170-172 : “We assumed that 20% of infections are asymptomatic and that asymptomatic people are as infectious as symptomatic individuals but missing the highly infectious pre-symptomatic phase”. This is an important hypothesis and it should be discussed.

  • We now emphasize in the discussion that this is an area of slight uncertainty in the model based on the ability of this estimate to change over time based on vaccination or new variants.

Throughout the manuscript: sometimes “within-host”, sometimes “intra-host” : please homogenize

  • These terms are interchangeable in the field but for this paper we now exclusively use intra-host.

Also please give exact result and not approximation as symbolized by “~”

  • We do not wish to include exact results because these simulations are intended as a tool to prove a scientific point rather than precise forecasts. Moreover, since the time of the simulations has not passed, precise numerical output carries little meaning. We hope that the reviewer can accept this perspective.

I also persist about adding legend elements in the main text: This is really disturbing the reading. If the reader needs the information while looking at the figure, then everything is already explained in the legend (the legends and captions are very well detailed). It is hence in addition redundant.

  • We agree and have removed this language.

Stating that the simulations represent variant B.1.1.7. seems too precise on the other hand. Since the difference between the “historical” and “B.1.1.7.” in the simulations result solely from a difference in infectivity, the authors should rather explain that they simulate “a variant” similar to the B.1.1.7. in terms of infectivity. In addition, the authors could explain that this approach would be easily applicable to other variants with other characteristics.

  • This is a fair point and we have made this change.

Line 414 : “With a vaccination rate of 5000 per day assuming the baseline variant, the number of 414 daily diagnosed cases was profoundly diminished…”. Could you please quantify “profoundly diminished”, please?

  • We now fill in this information.

In addition, provided the population size of King County, would  help the reader to better apprehend what the vaccination rates represent.

  • This is now provided

Line 485 : “estimate” should be replaced by “compute” for instance since this result was calculated from the “estimated reduction in number of infection” and not “estimated” itself. I am also not sure that this is an estimation since it results from simulation…

  • We have now used the term computed. Thank you.

Line 513 : “… this challenge remains relevant as new viral variants continue to emerge across the globe and these variants may result in lower vaccine efficacy.” Please provide a references

  • Provided

In the supplementary materials: please clearly state what M1 and M2 are. I thank you for explaining me what they were in your answer and I believe other readers would also be interested.

  • The model and these variables are not part of thew current model and are removed.

Figure S4: please increase font size.

  • This font is now increased

Reviewer 2 Report

I have attached a file with my comments.

Author Response

This paper integrates a within host model into a transmission system model in an informative manner. It presents both a useful model for modelers to build on and useful information from the model analysis for policy considerations. The main message at the start of the discussion section is a crucial message that must be quickly disseminated.

  • Thank you for this feedback.

But an adequate understanding the model is not conveyed to those who are likely to build on this methodology. One reason for the lack of clarity is that the description of the logic behind the within host model is left to another publication that the reader must chase down (Goyal et al, Science Advances). Then when the model in that paper is examined in detail, how the model in that paper generated the model in the submitted paper remains unclear. The supplementary material needs to do a better job of integrating the logic behind the intra-host model in lines 259-64.

  • We now add considerably more detail about the intra-host model in the supplementary information with a cartoon schematic of the model.

The intra-host model needs an explanation that does not require the non-modeler reader to go to the supplementary material. But it also needs a more explicit presentation of the model logic and formulation for modelers in the supplementary material. There is a disconnect between what is presented in the main text and what is presented in the supplementary material. Also needed is a table describing the variables and parameters. The integration of the intra-host model into the transmission system model is one of the great virtues of this paper. I hope the authors can quickly make the within host model more understandable.

  • To be clear, the intra-host modeling is separate from the epidemiologic model which was emphasized clearly in the last version. We now present more detail of the intra-host model as requested to clarify its equations, variables and parameter values. These are included in a supplementary table.

In the Intra-host model of SARS-CoV-2 kinetics section, is the delta meant to represent the infected cell density in the host? Define it formally and perform a dimensionality analysis to be sure that the definitions make sense within the model. How is delta measured? Below it is given dimensions that imply a rate per cell. What is discussed in the Supplementary material does not seem to correspond the model in lines 260-263 in the main text. The model presented in the main text lines 260-263 should be explained in the main text.

Should the “copies” in the V(0) definition be viable viruses, viable infected cells, or what?

  • This is viral genomic copies which is now stated in the revision.

There are no M1 and M2 in the lines 260—263 model. There is just an m.

  • Sorry, we have eliminated this line.

The “Intra-host” transmission section is describing “Inter-host” transmission, that is to say between host transmission. Intra-host (within host) transmission implies a cell-to-cell transmission process within an individual. The intra-host dynamics affect the Inter-host transmission in this model. That is the great virtue of this paper. But the description of this key aspect of the model seems to be lost in the main paper and in the supplementary material! I hope this can be corrected as soon as possible. It seems like these sections were written by different people who did not coordinate with each other and that the senior authors failed to insure this coordination.

  • This is not the case. I, Josh Schiffer, wrote and re-wrote the entire manuscript. I emphasize more clearly in the revised manuscript that the intra-host model is used for transmission modeling in sections 3.8 but is not at all integrated with the King County model throughout the rest of the paper.

Minor comments:

Page 3 Should “third” in line 78 be fourth as that is the fourth measure on the table of effects.

  • Thank you. We have changed this accordingly.

The new sentence in lines 92-95 is messed up and needs rewriting. Stop the first sentence on line 93. Then write another sentence relevant to that.

  • In our opinion, this sentence was clear as written but we have tried to make it more interpretable to this reviewer.

The sentence on lines 127-9 is also difficult to parse. The problem is “associated”. Are you trying to say that the secondary attack rate is 85% lower for asymptomatic than symptomatic infection? If not, what are you trying to say? This whole paragraph could be made clearer as to why the available data does not adequately provide the desired measure. Since the review has taken time, you might as well throw the delta variant into the last sentence before Materials and Methods.

  • Thank you. We agree with this critique. We have reworded to capture the fact that the secondary attack rate is 85% lower for asymptomatic than symptomatic infection

The sentence from lines 150-156 is unreadable. Break it up and add appropriate referential phrases.

  • We have tried to clarify this section.

Line 204: and then allowed it to vary prospectively

  • Changed accordingly

Before the equations in the supplementary material, it says that a series of compartments are presented in supplementary figure 1. That is not true. Such a figure is very much needed, along with a discussion of the logic for this series of compartments.

  • This is now included as another supplementary figure.